# A systematic review of *Leptospira* in water and soil environments

**Emilie Bierque**, **Roman Thibeaux, Dominique Girault, Marie-Estelle Soupé-Gilbert, Cyrille Goarant** *

Leptospirosis Research and Expertise Unit, Institut Pasteur in New Caledonia, Institut Pasteur International Network, Noumea, New Caledonia

* cgoarant@pasteur.nc

## Abstract

### Background

Leptospirosis, caused by pathogenic *Leptospira*, is a zoonosis of global distribution. This infectious disease is mainly transmitted by indirect exposure to urine of asymptomatic animals *via* the environment. As human cases generally occur after heavy rain, an emerging hypothesis suggests that rainfall re-suspend leptospires together with soil particles. Bacteria are then carried to surface water, where humans get exposed. It is currently assumed that pathogenic leptospires can survive in the environment but do not multiply. However, little is known on their capacity to survive in a soil and freshwater environment.

### Methods

We conducted a systematic review on *Leptospira* and leptospirosis in the environment in order to collect current knowledge on the lifestyle of *Leptospira* in soil and water. In total, 86 scientific articles retrieved from online databases or institutional libraries were included in this study.

### Principals findings/significance

This work identified evidence of survival of *Leptospira* in the environment but major gaps remain about the survival of virulent species associated with human and animal diseases. Studies providing quantitative data on *Leptospira* in soil and water are a very recent trend, but must be interpreted with caution because of the uncertainty in the species identification. Several studies mentioned the presence of *Leptospira* in soils more frequently than in waters, supporting the hypothesis of the soil habitat and dispersion of *Leptospira* with re-suspended soil particles during heavy rain. In a near future, the growing use of high through-put sequencing will offer new opportunities to improve our understanding of the habitat of *Leptospira* in the environment. This better insight into the risk of leptospirosis will allow implementing efficient control measures and prevention for the human and animal populations exposed.

**Data Availability Statement:** All relevant data are within the paper and its Supporting Information files.

**Funding:** The author(s) received no specific funding for this work.

**Competing interests:** The authors have declared that no competing interests exist.

# 1 Introduction

Pathogenic *Leptospira*, the etiological agents of leptospirosis, occur worldwide. This infectious disease affects people living in temperate and tropical climates in both rural and urban areas. Previous studies have estimated that the disease is responsible for at least 1 million cases and nearly 60,000 deaths annually [1]. This bacterial infection is frequently asymptomatic or initially presents as a flu-like febrile illness, making its clinical diagnosis challenging. Patients can then develop severe illness as Weil's disease (jaundice, bleeding and acute renal failure), and/ or severe pulmonary haemorrhage [2].

Pathogenic leptospires multiply in the renal tubules of chronically infected mammals [3]. Then, bacteria are shed via urine into the environment. Humans can be exposed directly or indirectly: veterinarians, farmers and meat workers, for example, may be in contact with infected kidneys or urine. However, indirect contamination through the environment is the most frequent human exposure route. This complex epidemiology makes it a paradigm of a One Health disease. Cases of leptospirosis occur after both occupational and recreational activities [4]. Transmission can occur by contact between wounded skin or mucosae and contaminated soil or water. Leptospirosis had long been known as an environment-borne infection, even before its etiological agent could be identified [5] and the term of "environmental reservoir" of leptospirosis has been proposed for soils in endemic regions [5–15]. Consequently, studies have focused on source investigations and on environmental risk factors to understand interspecies contaminations [16]. Besides risk factors of global significance, there is evidence that the risk assessment of leptospirosis transmission should take into account the geographical scale studied in order to evaluate locally relevant environmental and socioeconomic factors of human contamination [17].

It is currently assumed that pathogenic virulent leptospires are unable to multiply in the environment [18,19]. However, although the survival capacity of most species outside a host is not questionable, little is known on the environmental factors and determinants conditioning this survival [20]. The capacity of *Leptospira* to adapt to parameters such as osmolarity inside a host or in nature was also shown to be species-specific and related to the size of the *Leptospira* genome [21]. Knowledge on the lifestyle and the survival mechanisms of pathogenic leptospires in the environment remains scarce. This contributes to our insufficient understanding of basic aspects of leptospirosis epidemiology. More precisely, the capacity of different strains to survive in environmental conditions remains largely unexplored. Yet, understanding *Leptospira* survival is of prime importance for a better control and prevention of human leptospirosis.

Generally, massive leptospirosis outbreaks occur after heavy rain or flooding, notably after storms or hurricanes. Such outbreaks have been described in many tropical countries such as Brazil [22], Nicaragua [23], Sri Lanka [24] or the Philippines [25] among others, but seasonal peaks exist in most regions including tropical islands [26–30], illustrating the numerous environmental drivers of this disease [31]. Consequently, global climate change is expected to have an influence on the incidence and the distribution of leptospirosis [32–34].

This study aims to provide a systematic overview of the knowledge available on *Leptospira* presence and persistence in soil and water environments, including isolation and detection methods through a systematic literature review.

# 2 Materials & methods

## 2.1 Databases and search strategy

Articles were sought in April 2017 and the searches were further updated until December 2018 from three international databases: Medline (through PubMed), Scopus, and ScienceDirect.

We used a combination of the following search terms: [Leptospir* AND (soil OR water OR mud OR ecology OR hydric OR telluric OR environment OR paddy)]. The search terms were chosen to account for the diversity of words used to describe the soil and water ecosystems. In addition, the search term "ecology" was included to capture articles dealing with *Leptospira* ecology more generally.

### 2.2 Article selection process

The stepwise selection of articles was based on the strategy presented in Fig 1. First, duplicate articles were identified by sorting article titles in alphabetical and publication date order using a custom formatting of the database on an Excel spreadsheet. An additional identification of duplicates was made manually. Duplicates were discarded. Secondly, only articles in English or French were kept for further consideration. The third step consisted in excluding references that did not correspond to original scientific articles (e.g. indexes, abstracts, meeting announcements, reviews, posters, course material). In addition, articles that were out of the scope of this systematic review (e.g. about *Leptospirillum*, dealing with other viral or bacterial diseases, toxicology) were also excluded. Similarly, articles dealing with leptospirosis but without any link with the environmental aspect of the disease were removed. At this step, the availability of abstracts associated with articles was checked. Each abstract available was read by at least 2 researchers to confirm its relevance to the scope of the review and further proceed to the reading of the full length article. If a disagreement was observed, a third researcher was involved to further include or exclude the article. The full texts of the articles included for consideration in this systematic review were retrieved from various sources, including paper copies from institutional libraries. Articles whose abstract were not available were also read independently by 2 or 3 researchers to decide on their inclusion in the final analysis. The process is summarized in Fig 1.

### 2.3 Analysis of article content—Inclusion criteria

The full texts of all included articles were read independently by 2 or more researchers as described above, who collected all relevant data. This included the methods used as well as qualitative and/or quantitative results and taxonomic position of the strain as well as any other information useful for the understanding of *Leptospira* environmental survival. In addition, critical analysis of the articles allowed taking note of possible biases or limitations, including taxonomic uncertainties. Data was collected on Microsoft Excel for each individual article by one researcher and systematically checked by two other contributors.

## 3 Results

### 3.1 Studies selected

The initial search on databases retrieved 10,884 articles in total using Scopus, Medline and ScienceDirect databases. After removing 1,401 duplicates, 9,021 articles, either in English or in French, were submitted to inclusion criteria. Then 7,381 original scientific articles were sorted according to the scope of our systematic review. Finally 410 articles dealt with *Leptospira* or leptospirosis with mention of possible study in the environment. Of these, 75 were selected based on the abstract and 11 based on their full text, leading to a total of 86 full-text articles included for our analysis, as summarized in Fig 1.

Despite the selection of relevant keywords and a rigorous selection process, our final selection included articles that did not provide any relevant information, mostly because *Leptospira* was not sought in the ecosystem or the authors failed to evidence its presence [35–38]. Another

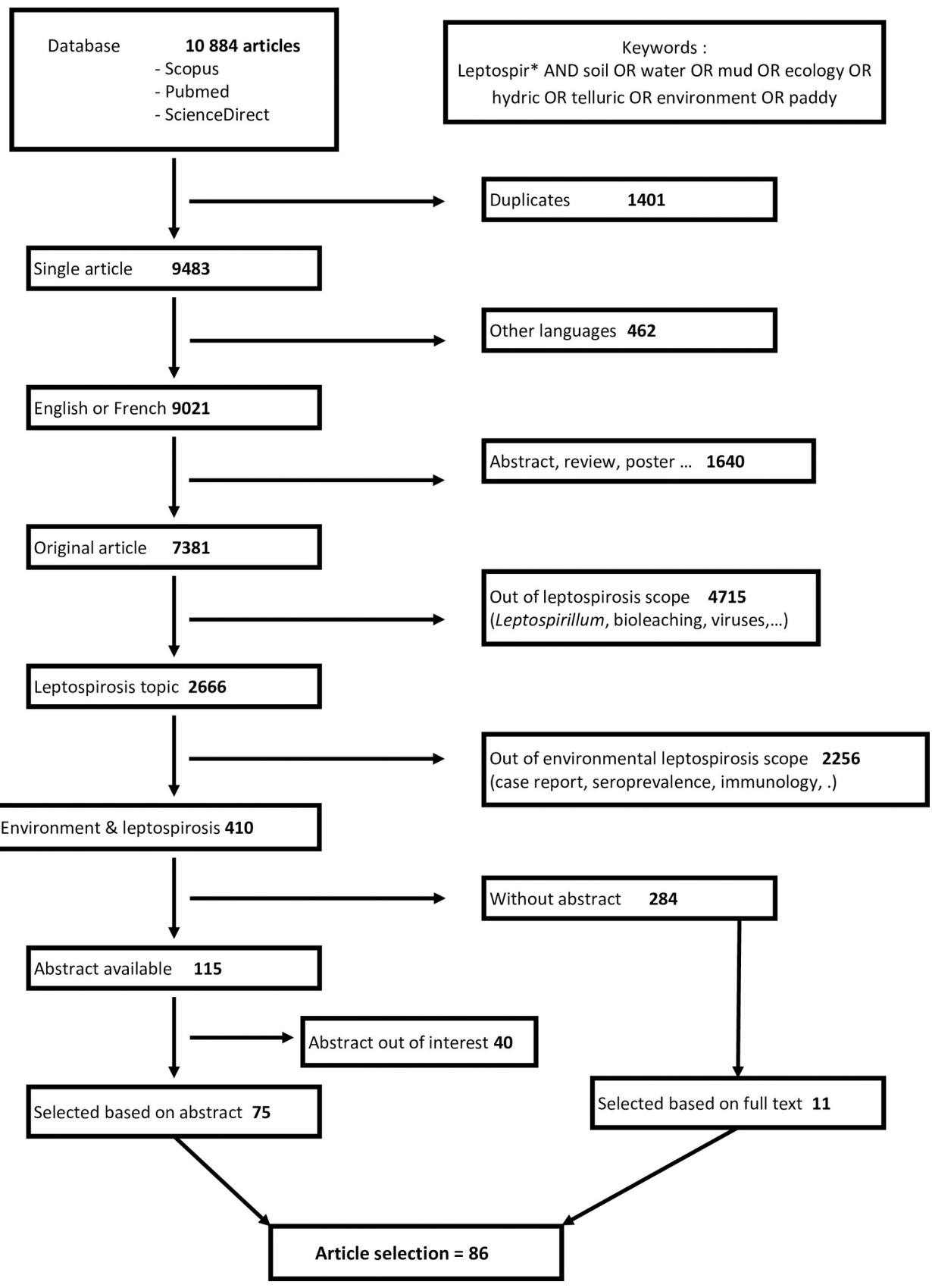

**Fig 1. Flow diagram of the systematic review and identification of the 78 articles included in our study.**

article fulfilling our selection process used former data to build a deterministic model, not providing original results [39]. In spite of these examples, our systematic review allowed collecting the techniques used and the current knowledge on *Leptospira* in the environment.

Factors usually reported to influence the survival of *Leptospira* in water or soils such as pH, salinity, temperature, moisture were not sufficiently reported in the studies selected to allow meta-analysis. However, it should be noted that a number of studies demonstrated *Leptospira* survival at low pH or low temperature in water or soils [11,40–45].

## 3.2 Methods used for the detection or isolation of *Leptospira* from the environment

**3.2.1 Molecular techniques to detect *Leptospira* in environmental samples and limitations.** Recent work suggests that some *Leptospira* within the "Pathogens" subclade have very low virulence towards mammals [14]. Oppositely, an increasing number of human leptospirosis cases have been reported as caused by *Leptospira* from the "Intermediate" subclade [46]. Together with comparative genomics data, this recently led to rename these subclades P1 and P2, with poor correlation to virulence in mammals [47]. Within the P1 subclade (formerly "pathogens"), another comparative genomics study separated 4 groups, namely Group I (*L. interrogans*, *L. kirschneri*, *L. noguchii*), Group II (*L. santarosai*, *L. borgpetersenii*, *L. weilii*, *L. alexanderi* as well as *L. mayottensis*), Group III (*L. alstonii*) and group IV (*L. kmetyi*) [48]. Within subclade P1 and to date, only species from Groups I and II have been isolated from humans or mammals and are considered virulent. Therefore, most articles reporting the molecular detection of "pathogenic" *Leptospira* must be interpreted with caution as they may not evidence the presence of virulent leptospires.

Techniques of leptospiral DNA amplification have been developed in order to detect leptospires from environmental samples. Early studies used 16S rRNA primers to detect saprophytic and pathogenic *Leptospira* [49,50] but most PCR techniques target genes only present in pathogenic *Leptospira* species like *lipL32* [51–55]. Other targets are sequences of *flaB* [10], *secY* [56–58] or *lfb1* genes [59,60] which display relevant polymorphisms; the PCR used and their possible applications to epidemiological studies have been reviewed recently [61]. Some studies combined two sets of primers to implement a multiplex PCR targeting both *lipL32* (a gene present only in species from the P1 and P2 subclades, but detected by most PCR only in species from the P1 subclade) and genus-specific *16S rRNA* (detecting all *Leptospira* spp.) to detect pathogenic *Leptospira* in environmental samples. The advent of real-time PCR has facilitated the acquisition of quantitative PCR data from environmental samples. Of note, there is also evidence that several targets for molecular detection of *Leptospira* in environmental samples can result in a high proportion of non-specific false-positive detections [62,63]. Even when specific, the detection of leptospires by PCR-based techniques does not provide any information concerning viability of cells in the environment. Yet, this point is of prime importance in the assessment of the risk of environmental transmission of the disease.

PCR methods have successfully been combined with the use of propidium monoazide to dramatically reduce the detection of dead or membrane-compromised cells. This technique, known as viability-PCR, provides indications about *Leptospira* viability in environmental samples [13,19]. Recently, some researchers have optimized procedures for the molecular detection of pathogenic leptospires from environmental waters [64], increasing possibilities for

further studies on environmental leptospirosis and opening avenues for real One Health studies of this complex zoonosis.

**3.2.2 Isolation of *Leptospira* from environmental samples.**   Isolating pathogenic leptospires from the environment is very challenging. However, some leptospires were historically isolated from environmental samples. Since saprophyte species, which are common inhabitants of the environment, are abundant and grow faster, they are the most frequently isolated from soil and water samples [8,65,66]. They also constitute a major difficulty to isolate virulent leptospires from Group I and II from surface water or soil by overgrowing these slow-growing strains [11,56]. Leptospires are sensitive bacteria in the laboratory. Strains from the P1 subclade are fastidious slow growing microorganisms with specific requirements [67]. The main culture media developed for *Leptospira* are the Korthof medium and the Ellinghausen McCullough Johnson Harris (EMJH) medium [68]. Different synthetic media have been developed for the culture of *Leptospira* [69], but EMJH is the principal medium used for both routine culture and isolation [70,71]. Benacer and colleagues [56] used it with addition of 5-fluorouracil to prevent contamination [72] while others used antibacterial and antifungal cocktails, the most recent being named STAFF (for sulfamethoxazole, trimethoprim, amphotericin B, fosfomycin and 5-fluorouracil) combination [6,10,11,14,73]. Another possibility to avoid contamination from water sample is the prefiltration through 0.22μm-pore size filters to inoculate culture media [10,74,75], although only a small proportion of leptospires pass through 0.22μm filter membranes [76]. This classical detection method does not allow quantification approaches because of a culture step. Likewise, this technique results in a loss of *Leptospira* diversity, notably accounting for the very rare isolation of significant pathogens from the groups I and II. Furthermore, isolation and culture techniques do not take into account possible viable but non-cultivable organisms, a physiological state never evidenced in *Leptospira*, but known from a large number of bacterial genera [77].

One of the historical methods used to isolate virulent leptospires has been the *in vivo* inoculation into susceptible animals. The strategy was to inoculate environmental samples (water or soil washings) directly into a susceptible animal in order to recover infecting leptospires in pure culture from blood. Still, this method does neither allow quantification, because pathogens are amplified or cleared by immune system of the host. Likewise, *Leptospira* infection can be concealed by another infection that kills the animal, preventing to isolate the virulent leptospires.

Lastly, Electron Microscopy studies provided evidence of *Leptospira*-shaped Spirochetes in microbial mats from salt marshes, but did not provide unambiguous evidence that these organism belong to the genus *Leptospira* [78].

## 3.3 *Leptospira* in water environments

**3.3.1 Occurrence of *Leptospira* in water.**   Studies have identified DNA sequences of *Leptospira* from the P1 subclade in drinking water (for human or for animals) samples [79–82]. This suggested a significant health concern and opened the way to consider *Leptospira* in studies on the potential risks associated with drinking water [83,84]. In 2017, Zhang and colleagues used metagenomics approaches to get insight into microbial communities of an urban drinking water system. Different pathogenic bacteria genomes were found in their dataset. An almost complete *Leptospira* genome was also retrieved; however, a Multi Locus Sequence Typing analysis shows that it corresponded to a saprophytic species [85]. A study in Colombia investigated the presence of *Leptospira* from the pathogenic subclade in drinking water systems and detected DNA of *Leptospira* from the P1 subclade in 41% of water fountains in Cali [86].

Environmental freshwater is one of the main sources of leptospirosis for humans and animals. In this typical One Health context, early studies attempted to identify potential human contamination sources [87–93]. Most often, isolation and identification of pathogenic *Leptospira* from surface water were attempted after contamination events [94] and data on occurrence were analyzed to explain past outbreaks [95]. As discussed above, authors who used culture-based methods mostly detected saprophytic strains [70,74,96–99]. When *Leptospira* from the Pathogenic subclade P1 were successfully isolated, they were most frequently related to the low-virulent Groups III & IV [8,65] or could not be further characterized [93]. In total, only very few studies described successful isolation of pathogenic leptospires from Group I and II, with proven virulence, from freshwater or soil [100–103]. Susceptible animal inoculation works provided isolates of virulent leptospires from creek's water samples in the USA [100] and from soils and water samples in Malaysia [101]. This technique however is currently unacceptable for ethical reasons because of the important number of susceptible animal used (almost 14,000 in the study in Malaysia [101]). More recently, *Leptospira interrogans* was successfully isolated from paddy water in Korea after inoculation into guinea pigs [102]. In Iowa (USA) scientists found the pathogenic "*Leptospira pomona*" (obsolete nomenclature, a pathogenic *Leptospira* from serogroup Pomona) during several years in surface waters used for recreational activities in the 1960s [100,104]. Later, molecular detections have facilitated studies. In Malaysia, where leptospirosis is endemic, leptospires from the P1 subclade have been found in up to 23.1% of lakes and recreational areas [56,75,105]. Similarly, Tansuphasiri detected leptospires from the P1 subclade in 23% of surface waters in Thailand [91]. These studies used classical PCR detection so did not collect quantitative data.

Only few studies provided quantitative data of *Leptospira* in environmental samples, mostly using quantitative real time PCR techniques. Estimations of concentrations of leptospires in surface water samples in Peru have evidenced from 1 to 17,147 leptospires per mL [106]. These authors demonstrated higher frequency but also higher concentrations in the urban area than in the surrounding rural areas. However, the PCR used also detected *Leptospira* from the P2 subclade [12] and sequences mostly pointed to an unknown subclade of *Leptospira* spp. with no known virulence [106]. Recently, quantitative detection of pathogenic *Leptospira* have been conducted in France, a temperate region. From 47 water samples, 3 were positive with concentrations from $10^3$ to $10^4$ genome-equivalent per mL [51]. In subtropical climate, 98.8% of Hawaiian streams revealed the presence of *Leptospira* (from the P2 subclade) with concentrations between 5 and 1000 genomes per 100 mL; this study highlighted a strong correlation between *Leptospira* concentration in water and the measured turbidity [15]. Other turbid water sources are represented by sewage water, historically linked with human contamination among sewage-workers [5,107,108]. A recent study in a Brazilian urban slum has shown that pathogenic *Leptospira* DNA was detected in 36% of sewage samples and even more frequently during the rainy season, with a mean concentration of 152 bacteria per mL [7].

**3.3.2 Survival and persistence of *Leptospira* in water.** Table 1 presents findings of studies on the persistence or survival of *Leptospira* spp. in water. Noguchi was the first to demonstrate the survival of pathogenic *Leptospira* for up to one week in drinking water, already pointing to environmental survival as an important clue in the epidemiology of leptospirosis, probably pioneering the One Health concept for leptospirosis [109]. However, technical difficulties to culture and identify leptospires from environmental samples restrict our knowledge of the environmental survival of pathogenic leptospires.

Microcosms or larger mesocosms have been largely used to study the survival of virulent *Leptospira* under different physicochemical conditions (See Table 1). Studies have demonstrated their capacity to survive in soil and water for prolonged periods. In a study, the survival and virulence was maintained for more than 40 days in soil and more than 20 days in water

**Table 1. Studies about persistence of pathogenic Leptospira in the environment [11,13,19,40–45,67,109,116,119,128–132].**

| Matrix | Microorganism* | Survival (Days unless stated otherwise) or DNA persistence | Experimental Conditions | Geographical Area of Study, Country | Reference |
|---|---|---|---|---|---|
| East river water | "Leptospira icterohaemorrhagiae" | no survival | survival observed in (1) sample without treatment, (2) autoclaved sample and (3) filtered sample | laboratory experiment, USA | Noguchi 1918 |
| sewage water | | | | | |
| stagnant water | | | | | |
| horse stool emulsion | | | | | |
| sewer filtrate | | | | | |
| drinking water | | one week (infectious) | culture | | |
| non sterile distilled water with few large motile bacilli | Pathogenic *Leptospira* strain Flanders | 3 days | Flanders strain cultured 22 days in rabbit serum+Ringer's solution and then placed in non sterile distilled water with few large motile bacilli | | |
| Plain tap water with air contamination | "Leptospira icterohaemorrhagiae" | 18–20 | Inoculation 10^6 washed leptospires / mL of fluid 190 ml of water seeded with 10 ml of leptospiral suspension at 2.10^7 cells/ mL incubated at 25–27°C | laboratory experiment, USA | Chang et al. 1948 |
| Sterile tap water | | From 28 hours to 32 days depending on pH values | | | |
| Sterile tap water with 1% serum | | 98–102 | | | |
| Sterile tap water with 0.1% tryptose | | 50–54 | | | |
| Tap water with bacterial flora | | 10–12 | | | |
| Tap water with bacterial flora and 0.1% tryptose | | 36–40 hours | | | |
| Charles River water | | 5–6 | | | |
| Sea water | | 18–20 hours | | | |
| Domestic sewage undiluted | | 12–14 hours | | | |
| Undiluted sewage aerated | | 2–3 | | | |
| 10% sewage in tap water | | 3–4 | | | |
| 1% sewage in tap water | | 7–8 | | | |
| Sterile tap water 5–6°C | | 16–18 | Inoculation 10^6 washed leptospires / mL of fluid 190 ml of water seeded with 10 ml of leptospiral suspension at 2.10^7 cells/ mL incubated at different temperatures | | |
| Sterile tap water 25–27°C | | 30–32 | | | |
| Sterile tap water 31–33°C | | 26–28 | | | |
| Charles River water 5–6°C | | 8–9 | | | |
| Charles River water 25–27°C | | 5–6 | | | |
| Charles River water 31–32°C | | 3–4 | | | |
| 10% sewage in tap water 5–6°C | | 6–7 | | | |
| 10% sewage in tap water 25–27°C | | 3–4 | | | |
| 10% sewage in tap water 31–34°C | | 2–3 | | | |
| Soil from a sugarcane farm on an alluvial flat bordering a river with addition of rainwater "to a fully moist condition" Soil pH reported to be 6.1–6.2 | "L. australis A" | 15 (2/2 replicates) to 43 (1/6) 15 (5/ 5 replicates) | Soil inoculated with cultures, then Soil inoculated with the urine of an experimentally infected rat Detection by re-isolation in guinea pigs | Laboratory experiment, Queensland, Australia | Smith and Self 1955 |

*(Continued)*

**Table 1.** (Continued)

| Matrix | Microorganism* | Survival (Days unless stated otherwise) or DNA persistence | Experimental Conditions | Geographical Area of Study, Country | Reference |
|---|---|---|---|---|---|
| distilled water—pH 6–34 to 36°C | "*Leptospira pomona*" | motility 2 culture 1 | water inoculated with 2.10^6 leptospires/mL soil inoculated with 10^6 leptospires/2 gram microcosm assessment motility by observation on darkfield microscope and 0.1 mL for culturing | Laboratory experiment, USA | Okazaki and Ringen 1957 |
| distilled water—pH 6–20 to 26°C | | motility 11.3 culture 4 | | | |
| distilled water—pH 6–7 to 10°C | | motility 12.2 culture 8 | | | |
| distilled water—pH 6–2 to -2°C | | motility 0.23 culture 0.96 | | | |
| distilled water—pH 6 - -20°C | | motility 0.08 culture 0.04 | | | |
| distilled water—pH 7.2–34 to 36°C | | motility 6.8 culture 6.5 | | | |
| distilled water—pH 7.2–20 to 26°C | | motility 34.8 culture 29 | | | |
| distilled water—pH 7.2–7 to 10°C | | motility 54 culture 44.5 | | | |
| distilled water—pH 7.2–2 to -2°C | | motility 0.92 culture 1.35 | | | |
| distilled water—pH 7.2 - -20°C | | motility 0.8 culture 0.8 | | | |
| distilled water—pH 8.4–34 to 36°C | | motility 2.4 culture 2.0 | | | |
| distilled water—pH 8.4–20 to 26°C | | motility 17 culture 15 | | | |
| distilled water—pH 8.4–7 to 10°C | | motility 2.6 culture 2 | | | |
| distilled water—pH 8.4–2 to -2°C | | motility 0.42 culture 1.35 | | | |
| distilled water—pH 8.4 - -20°C | | motility 0.08 culture 0.08 | | | |
| Palouse river water (Washington, USA) | | motility 8 infection 10 | | | |
| filtered Palouse river water | | motility 99 culture 94 infection>18 | | | |
| autoclaved Palouse river water | | motility 47 culture 27 | | | |
| double-distilled water | | motility 18 infection 9 | | | |
| Dry soil | | motility 0 culture 2 hours | | | |
| Damp soil | | motility 3 culture 5 | | | |
| Water-supersaturated soil | | motility 193 culture 183 | | | |
| Phosphate-buffered distilled water at varying pH | 4 different pathogenic *Leptospira* | Strain-dependent effect of pH.From ~10 days at low pH (<6.3) to >100 days. | Inoculation of phosphate-buffered distilled water tubes with an unknown number of leptospires. Survival assessed by microscopic observation of motile organisms.Of note, the cells are not washed, so diluted culture medium is also seeded in test tubes. | Laboratory experiment, London, UK | Smith and Turner 1961 |
| Paddy field Water and artificcially inoculated water | Pathogenic *Leptospira* (serogroup Australis) | In paddy field: survived up to 7 days. In laboratory experiments: survived 3h at 42°C; 7 days at 0°C and 14 days at 30°C | initial innoculum : 0.1 ml of one week old culture Paddy water were autoclaved, innoculated and distributed in 2mL ampoules dropped back into the paddy rice field or incubated into water baths/incubators/refrigerated room at various temperatures | Taiwan, 1965 | Ryu and Liu 1966 |
| | Saprophytic *Leptospira* (serogroup Semaranga) | in paddy field: survived up to 7 days. In laboratory experiments: survived 6h at 42°C; 7 days at 0°C and 14 days at 30°C | | | |

(*Continued*)

**Table 1.** (Continued)

| Matrix | Microorganism* | Survival (Days unless stated otherwise) or DNA persistence | Experimental Conditions | Geographical Area of Study, Country | Reference |
|---|---|---|---|---|---|
| Soil 1—pH 5.3—Dry matter (DM) 9.5% | Pathogenic *Leptospira* serogroups Grippotyphosa, Hebdomadis, Sejroe | 6 hours | Soils inoculated with urine (0.5–0.8 mL) of Leptospira-carrying voles at ~4.10^6 leptospires/mL. 19 tests : 11 soil samples with different vegetation covers, pH and moisture. Survival determined by collecting twice a day, several mg of soil, resuspending in saline and examining by dark field microscopy | Laboratory experiment, Lake Nero, Yaroslav region, Russie, June-August 1970 | Karaseva et al 1973 |
| Soil 1—pH 5.5—DM 14.2% | | 8 hours | | | |
| Soil 1—pH 6.2—DM 16.5% | | 12 hours | | | |
| Soil 2—pH 7.1—DM 41.4% | | 3 | | | |
| Soil 2—pH 7.4—DM 49.7% | | 5 | | | |
| Soil 2—pH 6.8—DM 52.4% | | 5 | | | |
| Soil 2—pH 7.5—DM 65.4% | | 7 | | | |
| Soil 3—pH 6.9—DM 69.8% | | 14 | | | |
| Soil 3—pH7.4—DM 72.6% | | 14 | | | |
| Soil 3—pH 7.5—DM 74.3% | | 15 | | | |
| Soil 3—pH 6.5—DM 77.4% | | 15 | | | |
| Ringer's solution at pH 7.15 and 20 C | "*L. autumnalis*" Akiyami A | >30 hours | $10^5$ leptospires per ml : 1-ml of this suspension added to 100 ml of a buffered-test solution to obtain 990 leptospires/mL. Incubation was at 30 C for up to 17 days. | Laboratory experiments, North Carolina | Schiemann 1973 |
| buffered (5.33 mM phosphate) thiosulfate (4.95 mM) solution at pH 7.39 and 20°C | | >95 hours | | | |
| buffered (5.33 mM phosphate) thiosulfate (4.95 mM) solutions at pH 7.40 and 25°C | | ~120 hours | | | |
| buffered (5.33 mM phosphate) thiosulfate (4.95 mM) solutions at pH 7.40 and 30°C | | 75 hours | | | |
| buffered (10 mm phosphate) thiosulfate (4.95 mM) solutions at 20°C and pH 8.22/7.82 | | >80 hours | | | |
| buffered (10 mm phosphate) thiosulfate (4.95 mM) solutions at 20°C and pH 7.42/7.37 | | >80 hours | | | |
| buffered (10 mm phosphate) thiosulfate (4.95 mM) solutions at 20°C and pH 6.79/6.72 | | 25 hours | | | |
| EMJH medium at 37°C | 23 pathogenic *Leptospira* and *Leptospira biflexa* | 7–42 strain-dependent | 1-mL inocula initially. Cells were cultured in EMJH medium at 37°C on successive subculture at 7-days intervals. | Laboratory experiments, USA | Ellinghausen 1973 |
| phosphate-buffered solution with / without 1% Bovine Albumin | 14 pathogenic *Leptospira* strains | 7 days for all strains and conditions. Strain-dependent, but better survival with 1% Bovine Albumin compared to buffer alone. | 1-mL inocula initially. Cells suspensions were stored at 23–25°C for 7 days before assessment of viability of serial dilutions | | |
| Sandy loam acidic soil | Pathogenic *Leptospira* serogroup Pomona | Detection of live and virulent *Leptospira* up to 42 days | Soil incubated with 5x10^8 organisms. 10 g dried soil (40°C, 3 days) saturated at 75%, 100% and 125% water level incubated for 1, 3, 6, 10, 15, 22, 31, 42, 49, 56, 63 and 70 days. Each sample is, then, treated by added 20 mL sterile distilled water and agitated 4 hours before being centrifuged for 5 min at 3000 g. Soil washed supernatant. Culture and hamster inoculation. | Laboratory experiment, New Zealand | Hellstrom and Marshall 1978 |
| Sensitivity to UV in diluted culture broth | *L. biflexa* serovar patoc Patoc I | *L. biflexa* more resistant to UV than *L. interrogans* serovar Pomona | 2.10^6 leptospires/mL initially. 3-mL of cell suspension were exposed to UV radiation under red light. The UV radiation dose was varied by changing the time of exposure with an intensity of 2 J/ m2/s. Survival assessed by culture | Laboratory experiments, North Carolina | Stamm and Charon 1988 |
| | *L. interrogans* serovar Pomona | | | | |

(*Continued*)

**Table 1.** (*Continued*)

| Matrix | Microorganism* | Survival (Days unless stated otherwise) or DNA persistence | Experimental Conditions | Geographical Area of Study, Country | Reference |
|---|---|---|---|---|---|
| pH-buffered solutions (2.2–7.9) at different temperatures (25–50°C) | *L. interrogans* Icterohaemorrhagiae M-20 | Survival assessed over 4 days Survival depends on both pH and temperature (modelled in the article) *Leptospira* total mortality at temperatures ≥45°C | ~3x105 leptospires / 100μL initially. final survival assessed by re-culture factorial design investigating the combined effects of temperature and pH on survival. Analysis with a logistic regression model | Laboratory experiment, USA | Parker and Walker 2011 |
| Rain puddle | Pathogenic and Intermediate *Leptospira* | ~150 *Of note, a real field study, new Leptospira contaminations may have occurred* | soil sample 3-cm deep (7.8% moisture content) in a rain puddle Re-detection of the same isolate 5 months after the first sampling (same PFGE profile) | Fukuoka, Japan | Saito et al. 2013 |
| pH 5.65; 25°C | pathogenic *Leptospira spp L. interrogans* and *L. kirschneri* | 12 weeks | Rice field water and pond water were autoclaved. 12 ml at 0.5 McFarland standard of a logarithmic phase culture Spiking was performed by centrifuging 12 mL of the adjusted culture, discarding the supernatant, and then resuspending with 12 mL of Rice field water and pond water | Laboratory experiment, Thailand, 2001–2006 | Stoddard et al, 2014 |
| pH 5.65; 30°C | | 12 weeks | | | |
| pH 5.65; 37°C | | 12 weeks | | | |
| pH unadjusted (6.95 to 7.79); 25°C | | 12 weeks | | | |
| pH unadjusted (6.95 to 7.79); 30°C | | 12 weeks | | | |
| pH unadjusted (6.95 to 7.79); 37°C | | 10 / 12 weeks | | | |
| pH 8.65; 25°C | | 12 weeks | | | |
| pH 8.65; 30°C | | 12 weeks | | | |
| pH 8.65; 37°C | | 8 / 12 weeks | | | |
| Mineral bottled waters (5 different) | *L. interrogans* Icterohaemorrhagiae Nantes 564 | Survival 28–593 days depending on water and temperature. Of note, survival observed at low pH and low temperature | Non-sterilized mineral bottled water inoculated with a virulent *Leptospira interrogans* isolate (6.6x10^5 /mL). Waters incubated at 4°C, 20°C or 30°C for up to 20 months. Survival was assessed by re-culturing in EMJH after filtration through 0.45μm filters. Temperature, pH, salinity and water composition considered independently. Microbial flora of the waters not considered. Of note, the cells are not washed, so diluted culture medium is also seeded in test tubes. | Laboratory experiment, France | Andre-Fontaine et al. 2015 |
| River soils in tropical island | *L. interrogans* Pyrogenes | >63 *Of note, a real field study, new* Leptospira *contaminations may have occurred* | Re-detection of the same isolate 4 months after infection. Virulent leptospires were viable in soil up to 9 weeks Soil samples were submitted to DNA extraction using the PowerSoil DNA Isolation kit and viability was assessed by viability qPCR from soil washings | Field experiments, New Caledonia | Thibeaux et al. 2017 |
| Mineral water pH 7.2 | *L. interrogans* serovar Manilae strain L495 | >42 | 2x10^9 at Day 0 / 2x10^7 at day 42 Late-logarithmic phase *Leptospira* grown in EMJH are harvested by centrifugation and resuspended in the same volume of mineral water, pH 7.2. Concentration is determined daily by direct count of mobile leptospires in Petroff-Hausser counting chamber, viability is determined by growth in standard EMJH. | Laboratory experiment, Paris, France | Hu et al. 2017 |
| sewage from New Haven, USA | *L. interrogans* serovar Copenhageni *L. biflexa* Patoc | DNA persistence *L. interrogans* 8 days *L. biflexa* >28 days | Spiking 40g or mL of matrix by $10^6$ cells/g or mL incuvated under dark condition at 29°C 1g or 1mL harvest at each time point Use of viability-PCR to assess survival | Laboratory experiments, USA | Casanovas-Massana et al. 2018 |
| bottle spring water | | Cell survival *L. interrogans* 28 days | | | |
| sandy loam soil from Brazil | | *L. interrogans* DNA persistence 21 days—cell survival 28 days *L. biflexa* 28 days | | | |
| loam soil from New Haven, USA | | DNA persistence *L. interrogans* 8 days *L. biflexa* >28 days | | | |

* Taxonomy as presented by the authors, may be obsolete for old articles

[41]. This study, with others, confirmed that pathogenic leptospires can survive and remain virulent for several weeks in the water and soil environment. Environmental survival capacities depend on the species and strains [44]. Despite their increased recognition in human infections, little is known on the epidemiology of *Leptospira* from the P2 subclade [46]. Within the P1 subclade, the paradigm *L. interrogans* has been the model species to describe the well-known One Health epidemiology. However, other species are most likely unable to survive outside a host for prolonged periods, as was shown for *Leptospira borgpetersenii* [110]. Lastly, other species from the P1 subclade described more recently were only found in the environment, mostly soils, as is the case for *Leptospira kmetyi* [111]. Some of these novel P1 *Leptospira* species were also unable to cause infection in animal models, even questioning their real need of an animal reservoir [14]. Recent studies have shown survival and conservation of virulence ability of *Leptospira interrogans* for around 20 months in mineral bottled water [40]. However, in this latter study, the bacteria were inoculated with their EMJH culture medium (10 mL into 1.5 liters), rather mimicking highly diluted culture medium. This data remains the longest survival reported in such conditions like cold, acidic and nutrient-poor conditions.

### 3.4 *Leptospira* in the soil environment

**3.4.1 Occurrence of *Leptospira* in soils.**   Again pioneering, Noguchi appears to be one of the first to consider, one century ago, soils as a possible environment where *Leptospira* could survive [109]. Culture methods have allowed isolation from only 1.1% of lake Nero soil samples in Russia [112]. Kingscote has revealed in 1970 a correlation between the bedrock of Southern Ontario and animal leptospirosis, but did not assess the presence of pathogenic *Leptospira in situ* [113]. Because of the technical challenges of studying delicate slow-growing organisms in soils, this environmental compartment remains poorly studied.

As discussed above, the advent of molecular techniques has allowed detection of *Leptospira* from soil samples. In Taiwan, 30.6% of farm soils that have been flooded sheltered pathogenic (P1) and non-pathogenic *Leptospira* [114]. The same proportion (31%) of *Leptospira* from the P1 subclade were found in soils of an urban slum in Brazil using *lipL32* qPCR [12]. In New Caledonia, the biodiversity of *Leptospira* isolated from soil has been revealed recently by identifying 12 novel species using MALDI-ToF mass spectrum and whole-genome sequencing analysis [6]. Using culture techniques and DNA detection, *Leptospira* has also been found in environmental samples from Malaysia [111,115]. In the same country, pathogenic *Leptospira* were recovered from soil washings in the early 1960s using animal inoculation. Interestingly, authors noted a higher isolation frequency from soil washings than from waters [101]. Quite similarly, a study in Minnesota showed a higher isolation frequency of saprophytic leptospires from soils than from adjacent waters [9]. Once again, in Hawaii, leptospires were isolated from 7 of 13 water samples, but from all 16 soil samples examined [99].

**3.4.2 Survival and persistence of *Leptospira* in soils.**   Table 1 also includes findings of studies on the persistence or survival of *Leptospira* spp. in soils. Okazaki and Ringen probably pioneered the field of soil microcosms to study *Leptospira* survival and evidenced a 6-month survival of a virulent *Leptospira* in water-saturated soil [116] (See Table 1). Another study confirmed survival and virulence after 6 weeks in soil microcosms [42]. Recent work has studied DNA persistence and viability of virulent leptospires in soil and water using microcosms [19] (see Table 1). In this work, the authors have used viability-PCR and shown a rapid decay of DNA in soil and sewage, allowing to assume viability from a direct qPCR from these matrices, contrasting with their findings in water where DNA detection does not demonstrate survival due to longer persistence of free DNA in water [19]. In this study, the authors built a model of *Leptospira* (both *L. interrogans* and *L. biflexa*) persistence in the soils, waters and sewage

studied, notably showing survival of *L. interrogans* up to 3 weeks in one soil together with a stronger survival capacity of the saprophyte *L. biflexa* [19].

In field conditions in New Caledonia, data have shown the capacity of *Leptospira interrogans* to survive and remain virulent in riverbank soils and sediments up to 9 weeks after human infection events [13]. Similarly, the putative same *L. alstonii* strain (identical PFGE profile) was isolated twice five months apart from the same soil in Fukuoka, Japan [11]. However, these findings can also be explained by repeated contaminations from animals or the surrounding environment and the persistence of virulent leptospires in real environmental conditions deserves further studies.

Together with survival studies, the increasing number of novel *Leptospira* isolated from the environment raises the question of their lifestyle. Recent studies have developed an interest in soil compositions affecting survival of *Leptospira*. Lall and collaborators have highlighted positive correlation between the presence of pathogenic leptospires and soil nutrients such as nitrate but also with metals as iron, manganese and copper. This work may help the comprehension of environmental transmission of the human and animal disease in a One Health approach [117] but it remains crucial to better understand the survival of pathogenic *Leptospira* in soil.

## 4 Discussion

Our systematic review used a rigorous process to identify the published literature on *Leptospira* survival and persistence in the environment in relation to leptospirosis. There is strong and convergent evidence that virulent leptospires can survive and remain infectious in the environment for months, notably in soils. However, no definitive proof could be obtained from field studies in open environments, where water and soils could possibly be exposed to repeated contamination from animals or the surrounding ecosystem. In addition, the current molecular tools hardly predict the true virulence of *Leptospira* upon molecular detection in the context of our changing comprehension of virulence in this complex bacterial genus [47].

Mesocosms and microcosms studies have been used in the past and have recently attracted renewed interest. Complementary approaches linking field studies and lab-controlled experimental evidence can help gain further insights into *Leptospira* environmental ecology and leptospirosis epidemiology.

Numerous studies have shown the consequences of heavy rain triggering massive outbreaks of leptospirosis. An emerging hypothetical mechanism is that virulent leptospires survive in soils and that rains wash soil surfaces, putting particles, including leptospires in suspension into surface water [12,13]. Thereby, leptospires would reach streams and freshwater bodies where humans get exposed. This hypothetical mechanism is depicted in Fig 2 and bibliographic data supporting this hypothesis are summarized in Table 2.

Studies have revealed higher isolation rates of *Leptospira* from soil than from freshwater samples [9,99,101]. These findings support the hypothesis that soils may be the original habitat of the genus *Leptospira* and a possible environmental reservoir or at least a temporary carrier of pathogenic strains [6,14,19]. This mechanism hypothesis is also supported by other findings, notably the positive correlation between *Leptospira* concentration in water and turbidity shown in Hawaiian streams [15]. The epidemiological records also suggest that human exposure occurs during the heavy rain events, or shortly after during floods, also supporting this hypothesis [118].

Little is known on survival strategies and physiological mechanisms used by leptospires [119]. However, strategies of positive interactions with environmental microbiota and biofilm formation are now well-known for other bacteria. Pathogenic leptospires have capabilities to

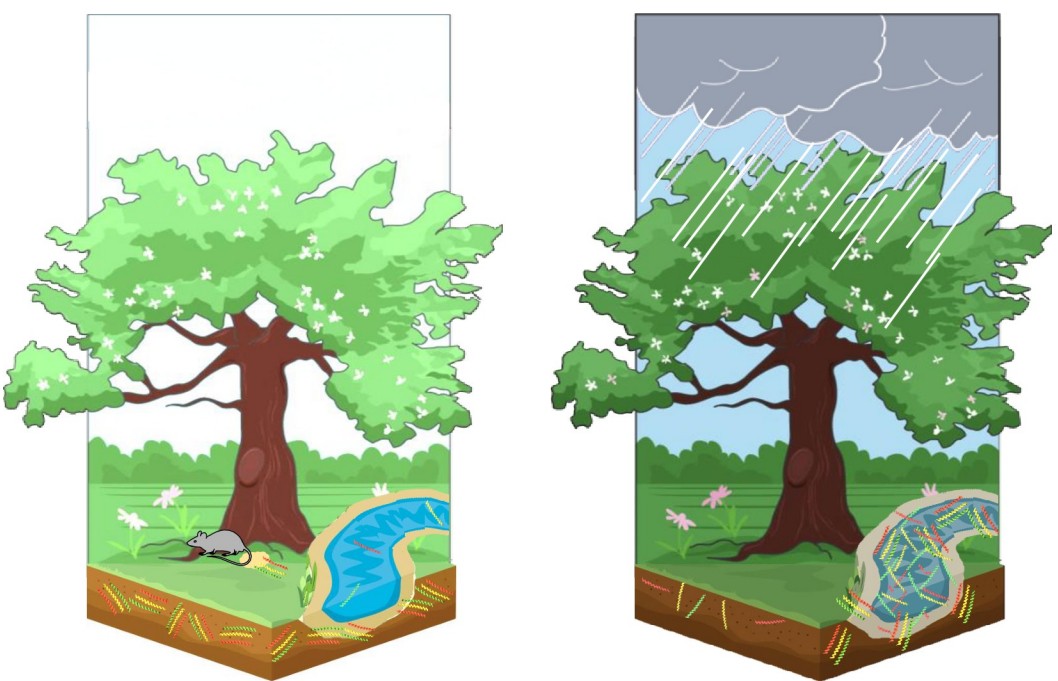

**Fig 2. Figure summarizing the hypothetical mechanisms of *Leptospira* environmental survival and dispersion upon heavy rainfall.** Table 2 identifies data supporting this hypothesis.

**Table 2. Significant findings supporting the hypothesis of the leptospiral dispersion from soil to water depicted in Fig 2.**

| Evidence supporting the hypothetical model | Geographical areas, Countries | Type of samples (% positive samples) | Reference |
|---|---|---|---|
| More frequent detection or isolation of *Leptospira spp*. from soils or sediment than from water | New Caledonia | Stream water vs sediment or bank soil (0% vs 57%) | [13] |
| | Malaysia | Stagnant water vs soil (19% vs 67%) | [115] |
| | Minesotta, USA | Lake shore water vs soil (65% vs 75%) | [9] |
| | | Bog water vs soil (5% vs 44%) | |
| | | Spring water vs soil (28% vs 59%) | |
| | Malaysia | Water vs soil (5% vs 18%) | [75] |
| | Hawaii | Water vs soil (54% vs 100%) | [99] |
| *Leptospira* concentration (log) has a significant positive correlation with turbidity (log) | Hawaii | Coastal stream water | [15] |
| Higher *Leptospira* concentration upon rainfall | Brazil | Surface waters | [7] |
| High concentration and genetic diversity of *Leptospira spp*. in soils, supporting the hypothesis of soils being a natural habitat of *Leptospira* spp. | New Caledonia | Soils | [6] |
| | Japan, New Caledonia, Malaysia | Water and soils | [47] |
| | Japan | Soils | [65] |
| | Philippines | Soils | [10] |
| | Brazil | Soils | [12] |
| Soils apparently protect *Leptospira* from seawater toxicity | Philippines | Soils | [10] |
| *Leptospira* survive in wet soil on dry days and appear in surface water on rainy days | Philippines, Japan | Soils and water | [11] |
| *Leptospira* concentration in surface waters correlates with rainfall intensity | Japan | River water | [127] |
| suggest that disturbance of river sediments increase the *Leptospira* concentration in water | | | |

form a biofilm *in vitro* as well as to survive in biofilms *in natura*, even in nutrient-free environments [120,121]. Genetic mechanisms underlying biofilm formation of *Leptospira* have been studied recently [122]. Furthermore, Vinod Kumar and collaborators highlighted cell coaggregation of pathogenic leptospires with other environmental bacteria within biofilms [53,123]. Supporting the protective role of biofilm [124], antibiotics tolerance was increased 5 to 6-fold by biofilm formation [125]. Altogether, these results support the hypothesis that *Leptospira* survival in the environment might be favored by biofilm formation or protection within a multispecies natural biofilm, but the precise interactions of virulent *Leptospira* in complex environmental microbiota remains to be determined.

*Leptospira* may also interact with other members of soil communities such as Free-Living Amoebas. Amoebas, which are one of the main colonizers of drinking water networks, are known to be possible reservoirs of potentially pathogenic bacteria. Diversity of cultivable amoebas and their bacterial community were analyzed by sampling a large drinking water network; *Leptospira* was found to be part of the bacterial community associated with Amoebas from surface water samples [79]. However, our review did not identify any study of the possible interaction of virulent leptospires with free-living Amoeba. This hypothesis would deserve to be considered through both field investigations and laboratory studies.

High throughput sequencing techniques are increasingly used in microbial ecology studies. These technologies allow identifying the diversity of bacterial communities. For instance, using modern molecular techniques, *Leptospira* reads were detected in Chinese river sediments receiving rural domestic wastewater [126]. Using shotgun sequencing (that offers higher sequencing depth) an almost complete genome of a saprophytic *Leptospira* sp. was retrieved from a drinking water network [85]. More recently, a study used environmental DNA metabarcoding and ecological techniques targeting *Leptospira* spp. and Vertebrates in Japan, successfully evidencing Group I *Leptospira* in the environment and providing an unprecedented insight into animal / *Leptospira* / weather ecological associations in a very elegant One Health approach [127]. These technologies and the corresponding datasets offer unique opportunities to gain new knowledge on *Leptospira* habitat in the water and soil environments.

Although the role of the environment in leptospirosis epidemiology was suspected more than a century ago [109], major knowledge gaps remain in our understanding of the survival and persistence of virulent *Leptospira* in the environment. The advent of environmental metagenomics and the combination of field studies with laboratory-controlled experiments are gaining renewed interest. This will offer new opportunities to better understand the environmental risk of leptospirosis and allow the implementation of efficient control measures.

## Supporting information

**S1 PRISMA checklist.**
(DOC)

## Acknowledgments

The authors would like to thank Noémie Baroux for developing Excel tools to identify rapidly duplicate articles.

## Author Contributions

**Conceptualization:** Cyrille Goarant.

**Data curation:** Emilie Bierque, Cyrille Goarant.

**Formal analysis:** Emilie Bierque, Roman Thibeaux, Dominique Girault, Marie-Estelle Soupé-Gilbert, Cyrille Goarant.

**Investigation:** Emilie Bierque, Roman Thibeaux, Dominique Girault, Marie-Estelle Soupé-Gilbert, Cyrille Goarant.

**Methodology:** Cyrille Goarant.

**Supervision:** Cyrille Goarant.

**Validation:** Emilie Bierque, Roman Thibeaux, Dominique Girault, Marie-Estelle Soupé-Gilbert, Cyrille Goarant.

**Visualization:** Emilie Bierque, Cyrille Goarant.

**Writing – original draft:** Emilie Bierque, Cyrille Goarant.

**Writing – review & editing:** Roman Thibeaux, Dominique Girault, Marie-Estelle Soupé-Gilbert, Cyrille Goarant.

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
