## [Decision Letter · Decision Letter 0]

2 Dec 2019

PONE-D-19-28972

A systematic review of Leptospira in water and soil environments

PLOS ONE

Dear Dr. Goarant,

Thank you for submitting your manuscript to PLOS ONE. After careful consideration, we feel that it has merit but does not fully meet PLOS ONE’s publication criteria as it currently stands. Therefore, we invite you to submit a revised version of the manuscript that addresses the points raised by reviewers #2 and #3.

We would appreciate receiving your revised manuscript by Jan 16 2020 11:59PM. To enhance the reproducibility of your results, we recommend that if applicable you deposit your laboratory protocols in protocols.io, where a protocol can be assigned its own identifier (DOI) such that it can be cited independently in the future. For instructions see: http://journals.plos.org/plosone/s/submission-guidelines#loc-laboratory-protocols

We look forward to receiving your revised manuscript.

Kind regards,

Odir A. Dellagostin

Academic Editor

PLOS ONE

Journal Requirements:

2.  Please include your tables as part of your main manuscript and remove the individual files. Please note that supplementary tables (should remain/ be uploaded) as separate "supporting information" files

Additional Editor Comments (if provided):

Reviewers' comments:

Reviewer's Responses to Questions

**Comments to the Author**

1. Is the manuscript technically sound, and do the data support the conclusions?

Reviewer #1: Yes

Reviewer #2: Yes

Reviewer #3: Yes

2. Has the statistical analysis been performed appropriately and rigorously? 

Reviewer #1: N/A

Reviewer #2: N/A

Reviewer #3: N/A

3. Have the authors made all data underlying the findings in their manuscript fully available?

Reviewer #1: Yes

Reviewer #2: Yes

Reviewer #3: Yes

4. Is the manuscript presented in an intelligible fashion and written in standard English?

Reviewer #1: Yes

Reviewer #2: Yes

Reviewer #3: Yes

5. Review Comments to the Author

Reviewer #1: This is a timely review of a little understood part of Leptospira spp. transmission and the role contaminated environments may play for the infection of animals and humans. I believe this will be of interest to the field and that it could stimulate much needed research in the area.

Reviewer #2: The manuscript entitled “A systematic review of Leptospira in water and soil environments” bring interesting information from evidence of survival of Leptospira ssp. the environment. The systematic review data are consistent and well presented in the manuscript. However some minor revisions are needed before the manuscript be considered for publication:

Material and Methods:

Line 85: Is it possible updates the articles until 2019?

Discussion:

- The authors report generally that all pathogenic Leptospira species can survive in the environment. I suggest that some Leptospira spp. that has been found to be in soil or water should be better specifically discussed, considering the species;

Figure 2 - I suggest deleting Figure 2, it's uninformative;

Table 1 - Table 1 shows different Leptospira spp. under various experimental conditions. The table is large and detailed; however, few conclusions were drawn from it.

Reviewer #3: The manuscript A systematic review of Leptospira in water and soil environments by Bierque et al. is a comprehensive review on a subject that has increasingly been studied in the recent years. Due to the One Health approach, leptospires of human and animal origin have been studied in comparison to environment. Nevertheless, several points remain to be better understood, mainly regarding pathogenic strains.

The manuscript is well written and methodology is adequate. Nevertheless, my main criticism is that very few associations with the disease on humans and animals were discussed. The context of One Health should be better considered, including animal reservoirs and hosts. Due to its substantial importance nowadays, I strongly suggest that authors explore the One Health framework, including a friendlier look at reservoirs and hosts.

Besides that, the article presents many speculations and hypotheses, and few certainties.

Regarding figure 2, I suggest to analyze some issues that are being illustrated and verify if they in fact occur:

1. When there is no rain, Leptospiras are mostly found in undersoil, not in soil.

2. When there is no rain, no Leptospiras are found in the water.

3. When there is rain, no Leptospiras are found in the soil.

6. PLOS authors have the option to publish the peer review history of their article (what does this mean?). If published, this will include your full peer review and any attached files.

Reviewer #1: No

Reviewer #2: Yes: Sérgio Jorge

Reviewer #3: No

---

## [Author Response · Author response to Decision Letter 0]

5 Dec 2019

The reponse to reviewers comments are included in a separate file.

---

## [Editor Report · Decision Letter 1]

12 Dec 2019

A systematic review of Leptospira in water and soil environments

PONE-D-19-28972R1

Dear Dr. Goarant,

We are pleased to inform you that your manuscript has been judged scientifically suitable for publication and will be formally accepted for publication once it complies with all outstanding technical requirements.

With kind regards,

Odir A. Dellagostin

Academic Editor

PLOS ONE
---

## [Editor Report · Acceptance letter]

13 Dec 2019

PONE-D-19-28972R1 

A systematic review of *Leptospira* in water and soil environments 

Dear Dr. Goarant:

I am pleased to inform you that your manuscript has been deemed suitable for publication in PLOS ONE. Congratulations! Your manuscript is now with our production department. 

With kind regards,

on behalf of

Dr. Odir A. Dellagostin 

Academic Editor

PLOS ONE